# Extending Multi-Pathway Human Health Risk Assessment from Regional to Country-Wide—A Case Study on Kuwait

Mohammad Munshed [1,2,*], Jesse Van Griensven Thé [1,2], Roydon Fraser [1], Bryan Matthews [2] and Ashraf Ramadan [3]

1. Department of Mechanical and Mechatronics Engineering, University of Waterloo, Waterloo, ON N2L 3G1, Canada
2. Lakes Environmental Software, Waterloo, ON N2L 3L3, Canada
3. Environmental Pollution and Climate Program, Environment & Life Sciences Research Center, Kuwait Institute for Scientific Research, P.O. Box 24885, Safat 13109, Kuwait
* Correspondence: mmunshed@uwaterloo.ca

**Abstract:** Air pollution has emerged as a pressing global issue in recent decades. While criteria pollutants and greenhouse gases contribute to the problem, this article explicitly addresses hazardous air pollutants (HAPs). This work estimates the country-wide cumulative human health impacts from exposure to HAPs. Kuwait is used as the case study due to data availability and non-fragmentation of data. At present, the evaluation of multi-pathway human health risks arising from exposure to HAPs is incomplete, as indirect pathways have not been considered. Furthermore, only a few HAPs, such as benzene, have established ambient air quality standards specifically intended to safeguard human health, leaving many HAPs unregulated. This study considers several pathways (both direct and indirect) and various environmental media (air, water, plants, soil, and animal tissue). The findings indicate that cumulative health risks in the coastal air quality zone are within acceptable limits but are notably higher when compared to the other air quality zones. For cancer risks, only the Ahmadi Hospital, with a cancer risk of $1.09 \times 10^{-5}$ for the resident adult exposure scenario, slightly exceeds the acceptable risk level of $1 \times 10^{-5}$. The proposed methodology integrates the results from a country-wide emissions inventory composed of different air quality zones, air dispersion and deposition modeling, multi-pathway transport-and-fate analysis, exposure quantification, and health risk and hazard characterization. It also extends and adapts EPA methodologies initially designed for hazardous waste combustion facilities to additional emission sources and provides a case study for a region seldom subjected to such human health risk assessments.

**Keywords:** air pollution; Kuwait; human health risks; cancer; non-cancer hazards; multi-pathway exposure; emissions inventory; air dispersion modeling; cumulative health impacts; risk driver analysis

## 1. Introduction

Kuwait, a country in the Middle East, has been selected as the country of study, rather than, for example, the United States, due to the unfragmented nature of the data and the smaller size of the country. Kuwait is situated in the northeastern region of the Arabian Peninsula, bordered by Saudi Arabia to the south and Iraq to the north. It spans an area of approximately 17,818 square kilometers, slightly smaller than New Jersey in land area, and is located at the geographical coordinates of 29.3759° N, 47.9774° E [1].

Kuwait is a country that heavily relies on its abundant oil and gas resources, which have led to the development of various industries such as petrochemical production, refining, power generation, crude steel production using electric arc furnaces, and desalination using natural gas and heavy oil.

As a result of various oil, gas, and industrial activities, numerous pollutants are emitted from different emission source types. These include point sources, such as stacks

associated with compressors, boilers, turbines, incinerators, flares, and exhaust vents related to glycol dehydration processes. In addition, area and volume sources represent other categories of emission sources. Area sources encompass fugitive emissions characterized by non-buoyant releases with negligible vertical extent, such as equipment leaks or landfill emissions. In contrast, volume sources involve non-point emissions with an initial vertical extent, including prescribed burns and marine vessel emissions.

Additionally, on-road and non-road mobile sources contribute to pollutant emissions. On-road mobile sources comprise motorcycles, passenger cars, light-duty trucks, buses, and heavy-duty trucks. Non-road mobile sources include agricultural equipment, aircraft jets, forklifts, and construction equipment, such as graders and backhoes.

Due to various industrial activities, oil and gas operations, traffic congestion, and frequent dust storms in Kuwait throughout the year [2,3], these factors lead to poor ambient air quality. Recent research indicates a decline in air quality across both urban and rural areas of Kuwait [4–6], which has subsequently been associated with increased mortality and morbidity rates [7–9].

Kuwait has three distinct air quality zones (AQZs): coastal, inland, and production, which are further divided into eleven subzones [10]. Each air quality zone and subzone presents a unique combination of emission source types, pollutants, and exposure scenarios that must be considered. The proposed approach employs the validated U.S. Environmental Protection Agency (EPA) Human Health Risk Assessment Protocol (HHRAP) risk assessment methodology [11] established in the US and Canada.

### 1.1. Literature Review, Objectives, and Additional Contributions

1.1.1. Literature Review

Exposure to hazardous air pollutants (HAPs), also known as air toxics, is associated with various health impacts [12], leading to short-term and long-term health problems. The specific health effects experienced by the population depend on the type, concentration, and mixture of air toxics, as well as the duration and frequency of exposure. For instance, respiratory issues may arise from exposure to methyl mercaptan, which can aggravate respiratory conditions [13], resulting in increased coughing, wheezing, and shortness of breath. Moreover, cardiovascular diseases, including atherosclerosis, have been linked to exposure to polycyclic aromatic hydrocarbons (PAHs) such as benzo(a)pyrene [14–17].

Neurological disorders may also be associated with exposure to air toxics, as pollutants like mercury and lead can cause cognitive impairments and memory loss [18], and in severe cases, conditions such as convulsions and coma [19–22]. Additionally, reproductive issues, including fertility problems [23], hormonal imbalances [24,25], and congenital disabilities [26], have been associated with exposure to certain air toxics, such as dioxins and some PAHs.

Lastly, long-term exposure to HAPs, such as benzene, formaldehyde, and 1,3-butadiene, can increase the risk of developing several types of cancer, including lung cancer [27–32] and lymphohematopoietic cancers [33]. It is important to note that vulnerable populations, including children, the elderly, and those with pre-existing health conditions, may be more susceptible to the adverse effects of air toxics. The HAPs summarized in Table 1 represent only a small fraction of air toxics released into the ambient environment from various industrial processes. These chemicals have been selected for their potency and the extensive availability of both cancer and non-cancer assessments in the literature.

To date, no country-wide studies have addressed the cumulative human health risks associated with exposure to hazardous air pollutants. Furthermore, no studies have yet considered multiple exposure (i.e., direct inhalation and scenario-relevant indirect pathways) in combination with numerous different sources (e.g., oil and gas operations; wastewater treatment plants; etc.). See Table 2 for a summary of the types of human health risk assessments that have and have not been conducted.

**Table 1.** Human Health Effects of Selected Hazardous Air Pollutants (HAPs).

| Hazardous Air Pollutants | Emission Source Type | Major Exposure Route | Organ/System Affected | Citation |
|---|---|---|---|---|
| Methyl mercaptan | Point, area, volume | Inhalation | Respiratory | [13] |
| Polycyclic aromatic hydrocarbons (PAHs) | Point, area, volume, on-road, and non-road mobile sources | Inhalation; oral | Cardiovascular | [14–17] |
| Mercury | Point, area, volume | Inhalation; oral | Central nervous and peripheral nervous systems | [18] |
| Lead | Point, area, volume | Inhalation; oral | Central nervous and peripheral nervous systems | [19–21] |
| Dioxins | Point, area, volume, on-road, and non-road mobile sources | Oral | Reproductive | [23–26] |
| Benzene | Point, area, volume, on-road, and non-road mobile sources | Inhalation; oral | Hematological and respiratory | [27–31] |
| Formaldehyde | Point, area, volume, on-road, and non-road mobile sources | Inhalation; oral | Hematological | [32] |
| 1,3-Butadiene | Point, area, volume, on-road, and non-road mobile sources | Inhalation | Hematological and immune | [33] |

**Table 2.** Summary of Human Health Risk Assessments in Literature.

| Multi-Source | Multi-Pathway | Cumulative | Citation |
|---|---|---|---|
| Yes | No | No | [34] 2020; [35] 2015; [36] 2012 |
| No | Yes | No | [37] 2022; [38] 2019; [39] 2009 |
| Yes | No | Yes | [40] 2022; [41] 2019; [42] 2015 |
| No | Yes | Yes | [12] 2023; [43] 2014; [44] 2010; [45] 2006 |
| Yes | Yes | Yes | This Work |

1.1.2. Objectives

The specific objectives of this research, using Kuwait as a case study, are as follows:

1. To implement a country-wide cumulative human health risk assessment incorporating multi-source and multi-pathway exposures.
2. To compare country-wide human health risk variability by region, for example, coastal versus inland.
3. To identify the chemical risk driver (defined by the authors as the dominant chemical of potential concern) for a particular sensitive receptor.
4. To identify if the direct or indirect pathway is the dominant pathway of risk for a particular sensitive receptor.

Though beyond the scope of this paper, these objectives support the long-term goals of targeted risk reduction and targeted mitigation resource allocations.

1.1.3. Additional Contributions

1. Extending and adapting EPA methodologies initially designed for hazardous waste combustion facilities to additional emission sources, including wastewater treatment plants, glycol dehydration units, but not including fugitives and mobile sources.
2. Providing a case study for a region of the world seldom subjected to such human health risk assessments.

Figure 1 presents the methodology framework used in this study.

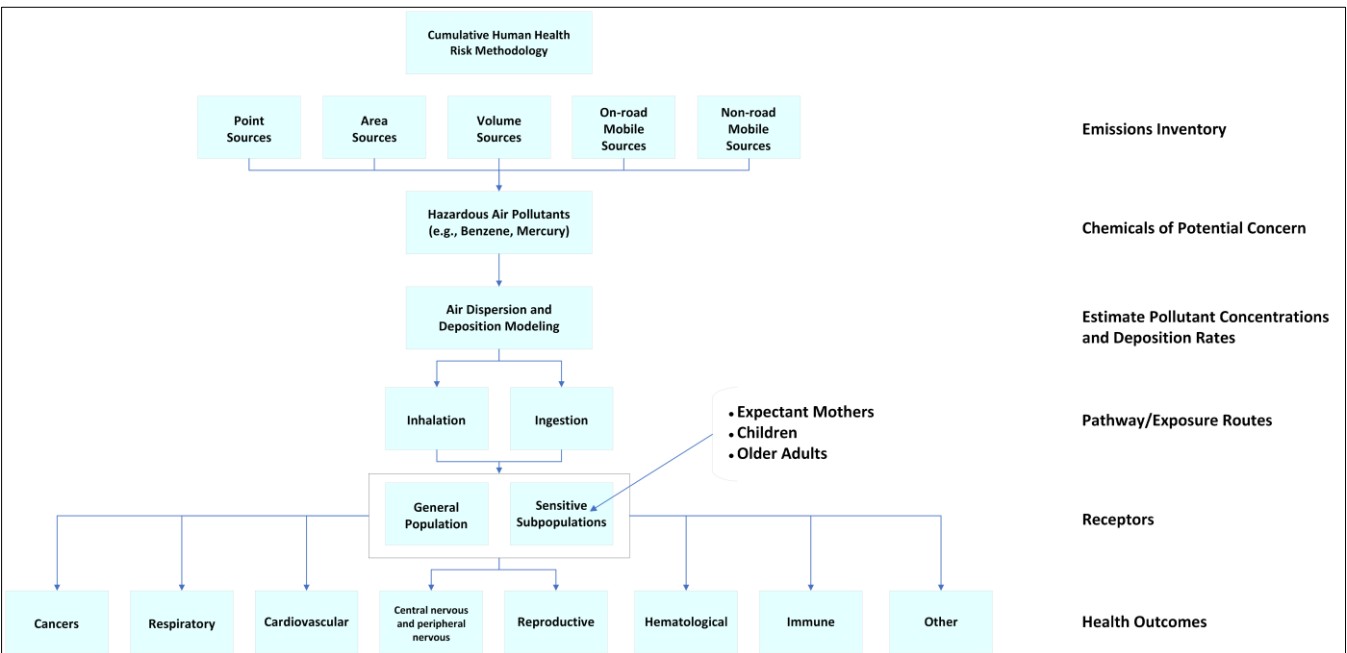

**Figure 1.** Flow diagram illustrating the application of a multi-pathway health risk assessment methodology in Kuwait's air quality zones.

## 2. Materials and Methods

The methodology consists of the following sequential steps:

1. Develop an emissions inventory (or augment an existing inventory) for each air quality zone.
2. Conduct air dispersion modeling using current US regulatory air dispersion models, such as AERMOD or similar, to estimate pollutant concentrations in the air and their deposition rates in various exposure media.
3. Estimate the concentrations of pollutants at the point-of-contact for receptor populations by conducting an environmental transport-and-fate analysis.
4. Identify realistic exposure scenarios to estimate the types and magnitudes of human exposure to pollutants.
5. Assess the levels, frequencies, and durations of contact between humans and pollutants.
6. Calculate multi-pathway and cumulative cancer risks and non-cancer hazards for each air quality zone (risk characterization).
7. Examine the contributing factors and underlying drivers of unacceptable risks (risk drivers).

### 2.1. Emissions Inventory (Step 1 in the Proposed Methodology)

An emissions inventory plays a vital role in managing air quality by enabling decisionmakers to:

1. Measure the contribution of each sector's emissions.
2. Analyze emission trends both retrospectively and prospectively.
3. Provide guidance and assistance to policymakers and industry in regulating emissions and establishing achievable targets.
4. Supply the necessary data for accurate assessment of current or future emissions using air quality modeling tools, with an accurate emissions inventory being a prerequisite for air quality modeling.
5. Identify locations for monitoring hazardous air pollutant hotspots.

In this study, emissions were calculated using the methods and algorithms summarized in Table 3 below.

**Table 3.** Overview of EPA and Industry Methods/Algorithms for Estimating Emissions from Various Sources.

| Method/Algorithm | Objective | Citation |
|---|---|---|
| EPA [1] AP-42 emission factors | Estimate emissions from various source categories, including point sources (e.g., industrial stacks) | [46] |
| AP-42, Fifth Edition, Volume I, Chapter 7: Liquid Storage Tanks | Calculate volatile organic compound (VOC) and hazardous air pollutant (HAP) emissions from floating- and fixed-roof storage tanks | [46] |
| EPA WATER9 | Estimate emissions from wastewater treatment plants | [47] |
| EPA LandGEM | Calculate emissions from landfills, utilizing the methane generation rate and potential methane generation capacity parameters | [48] |
| GRI-GLYCalc | Estimate emissions from glycol dehydration units | [49] |
| API [2] AMINECalc | Estimate emissions from amine gas treatment plants | [50] |

[1] U.S. Environmental Protection Agency. [2] American Petroleum Institute.

Table 3 lists methods chosen for their applicability, accuracy in emissions estimations, and broad acceptance within the scientific, regulatory, and industrial community. These validated methods were developed by reputable organizations such as the U.S. Environmental Protection Agency (EPA) and American Petroleum Institute (API), based on research, empirical data, and engineering principles.

For each air quality zone, unique identification codes were assigned to each emission unit and to each process based on the fuel type consumed. These codes facilitated the tracking of results and thorough examination of emissions from individual units and processes across facilities and sectors. This coding procedure was instrumental in the study's quality assurance process, enabling the review of assumptions, calculations, procedures, and the evaluation of the quality and representativeness of the inventory data [51].

It is important to note that emissions calculated using AP-42 emission factors were based on normal, steady-state conditions and did not take into account worst-case scenario conditions, such as during process start-up and shut-down, as well as process upsets. While these abnormal operating conditions have a relatively short time span, resulting emissions can easily exceed relevant emission limits and potentially cause ground-level concentrations that exceed relevant ambient air quality standards (e.g., National Ambient Air Quality Standards). According to Obaid et al. [52], various industry sectors, including power and/or heat generation, nuclear power generation, petrochemical production, energy-from-waste generation, and sulfuric acid production, exhibit different characteristics for start-up and shut-down emissions compared to steady-state conditions, and these characteristics differ for different source types and pollutants. The same authors note that attempts to characterize emissions during these conditions face numerous challenges, including the unavailability of manufacturers' design data and the complex relationship between different dynamic process functions and timing of events. In addition to these challenges, other uncertainties associated with the calculation of criteria pollutants and HAPs such as benzene, toluene, ethylbenzene, and xylene (BTEX) emissions include data gaps, data quality, and inconsistencies across different power plants and sometimes even different emission units within the same power plant.

The general formula for estimating emissions is [46]:

$$Emissions = A \times EF \times \left(1 - \frac{ER}{100}\right) \qquad (1)$$

where *A* represents the activity rate, *EF* is the emission factor, and *ER* is the total percentage (%) of emission reduction efficiency.

It is important to note that the availability and completeness of an emissions inventory are critical factors in evaluating the risks to human health associated with exposure to hazardous air pollutants.

### 2.2. Atmospheric Dispersion and Deposition Modeling (Step 2 in the Proposed Methodology)

Air dispersion modeling is defined as the mathematical representation of pollutant transport in the atmosphere, along with the quantification of deposition rates in various exposure media [12]. Air dispersion modeling is performed to understand the location and magnitude and to predict the spatial distribution of pollutant concentrations. It is also used to compare modeled and monitored air concentrations to evaluate the air dispersion model performance and representativeness of the emissions inventory. Moreover, air dispersion and deposition modeling is a prerequisite for conducting a multi-pathway human health risk assessment, as such modeling is used to estimate pollutant concentrations in the air, which are crucial for calculating inhalation cancer risks, and to estimate deposition rates, which are critical for assessing indirect exposure pathways.

In this study, air dispersion and deposition modeling was conducted using the current EPA-preferred regulatory model, American Meteorological Society/United States Environmental Protection Agency Regulatory Model (AERMOD) [53], employing version 22112.

AERMOD, an extensively validated steady-state Gaussian plume model, integrates air dispersion principles based on planetary boundary layer turbulence structure and scaling concepts. It addresses surface and elevated emission sources and accommodates simple and complex terrain. Since 2005, the EPA has approved AERMOD as its regulatory model, and it is widely used in various applications, including permitting, air quality assessments, and human health risk evaluations [53].

Designed as a near-field model for distances up to 50 km from the source, AERMOD exhibits robust performance across diverse terrains and under a wide array of meteorological conditions. AERMOD calculates concentrations using mathematical principles, such as plume rise, atmospheric dispersion of buoyant (or neutrally buoyant) effluent, turbulence theory, and Gaussian distribution [54].

The AERMOD modeling system does not utilize stability classes, which were used in the previous Industrial Source Complex Short Term 3 (ISCST3) model but employs a more advanced turbulence scaling parameterization scheme. The model predominantly relies on surface weather observations to generate vertical profiles of variables, including temperature, temperature gradient, wind speed, wind direction, and turbulent velocities within the atmospheric boundary layer. The EPA's AERMET model is the meteorological preprocessor responsible for producing AERMOD meteorological data files.

While validation of the AERMOD model is a desirable aspect of air dispersion modeling to ensure result credibility for a specific study area, such an evaluation was not conducted due to an absence of available measured air pollutant concentrations.

#### 2.2.1. Unitized Emission Approach

The unitized emission approach in air dispersion modeling is a highly efficient method for evaluating the impact of multiple pollutants emitted from a single source [55]. This technique relies on the principle that air concentration levels and deposition fluxes of pollutants are linearly related to the source's emission rate, meaning that a tenfold increase in emissions would result in a tenfold increase in receptor concentrations. By employing a standardized emission rate of 1 g per second (g/s) for the source, this approach streamlines the modeling process and significantly reduces computational time.

Using the unitized emission rate, a single air dispersion modeling run can provide concentration and deposition data for all emitted pollutants. This is achieved by multiplying the unit emission results by the actual emission rate of each individual pollutant. Consequently, the need for running the model separately for each pollutant is eliminated, enhancing the method's efficiency.

The output of this approach includes unitized annual average and maximum one-hour average air concentrations and deposition rates for the vapor, particle, and particle-bound phases. These unitized values can be further used as inputs for risk modeling, wherein media-specific concentrations and deposition of each contaminant are calculated by scaling the unitized concentrations by the appropriate pollutant emission rate.

The unitized emission approach is particularly beneficial for modeling a vast number of pollutants, as it minimizes calculation imprecision due to low emission values and simplifies the process by avoiding multiple runs for each modeled pollutant.

### 2.2.2. Meteorological Data

The Weather Research and Forecasting (WRF) model [12,56] was used to generate two AERMOD meteorological data files (surface and profile files) for each of the three air quality zones in Kuwait (i.e., coastal, inland, and production). To ensure adequate coverage across the country, three pseudo-meteorological stations (pseudo-stations) were strategically positioned. Pseudo-stations define the locations for which WRF data are generated, with meteorological parameters representative of the selected location.

### 2.2.3. Air Dispersion Model General Options

The modeling options and settings used in AERMOD for the analysis are provided in Table 4.

**Table 4.** AERMOD Model Options and Settings Used in This Study.

| Model Option/Setting | Setting |
| --- | --- |
| AERMOD Executable | Version 22112 |
| Dispersion Options | Non-default regulatory option selected<br>Fast all sources (FASTALL)<br>Flat (FLAT) |
| Calculation Type | Unitized (unit emission rate concept representing the $\mu g/m^3$ impact per 1 g/s of emissions) (refer to Section 2.2.1) |
| Output | Concentration, total deposition, dry deposition, and wet deposition |
| Dispersion Coefficient | Rural |
| Pollutants | Benzene, formaldehyde, toluene, and benzo(a)pyrene |
| Averaging Periods | 1 hour and annual |
| Source Types | Point |
| Receptors | Uniform Cartesian grid and discrete Cartesian receptors (sensitive receptors) |
| Terrain | Terrain in Kuwait can be approximated as flat; however, terrain files were used for completeness in the model |
| Meteorological Data Files | 2017 hourly meteorological data, contained in surface and profile files, for Kuwait's three air quality zones were generated utilizing the WRF model |

### 2.2.4. Receptors (Calculation Points)

Receptors are specific locations with designated coordinates where pollutant concentrations are calculated. These receptors can represent human populations, ecosystems, or other areas sensitive to air pollution. AERMOD uses meteorological data, emission source parameterizations, and terrain datasets, such as the National Aeronautics and Space Administration (NASA) Shuttle Radar Topography Mission (SRTM) digital files, to predict the dispersion and concentration of pollutants at these receptors, thereby helping to evaluate air quality and assess the potential impact on health or the environment.

In each of the three air quality zones, two uniform Cartesian receptor grids were established centered around the facility of interest. These grids were configured with both fine (100 m spaced grid from the centroid of the emission sources out to a radius of

3 km) and coarse (500 m spaced grid extending from 3 km to 10 km) settings to cover all areas where emissions may have a significant impact. The selection of receptor locations typically depends on regulatory requirements and the specific objectives of the air quality assessment, as different projects may have varying goals and criteria for receptor placement. Additionally, sensitive receptors, defined as locations where people are more susceptible to adverse effects from exposure to hazardous air pollutants (such as schools, hospitals, residential areas, daycare facilities, care facilities, and places of worship), were included in the project. Further information about these sensitive receptors can be found in the case study section of this paper.

### 2.3. Transport, Fate, Exposure, and Risk Characterization (TFER)

### 2.3.1. Transport-and-Fate Modeling (Step 3 in the Proposed Methodology)

This step examines the behavior and movement of hazardous air pollutants (HAPs) following their release into the atmosphere, which is critical for determining their potential impact on human health and the environment. Transport refers to the physical movement of HAPs as they travel through the environment, while fate pertains to their ultimate destination within environmental media (e.g., soil, food, water). This analysis aims to calculate the concentrations of HAPs in air, soil, produce, milk, meats, eggs, fish, and drinking water. The focus is on ensuring accuracy in assessing the presence of HAPs in different environmental media, which is vital for understanding their potential impact on human health and the environment. The media concentrations were estimated using over forty mathematical equations [11], including a system of non-linear equations, such as the cumulative soil concentration for carcinogens and non-carcinogens, and the hazardous air pollutants' loss constant due to runoff. The modeling approach accounted for numerous factors, such as meteorological conditions, physicochemical properties of the pollutants, and environmental characteristics of the study area, including annual evapotranspiration, annual irrigation, annual precipitation, and annual runoff. It is important to acknowledge that the modeling process may involve certain assumptions, limitations, or uncertainties that should be considered when interpreting the results. To facilitate transport-and-fate modeling, the computer application IRAP-h View [57] was used. An example of the output of the transport-and-fate modeling is listed below:

- Aboveground exposed produce concentration due to direct deposition: [mg/kg]

### 2.3.2. Exposure Quantification (Steps 4 and 5 in the Proposed Methodology)

Exposure evaluation is the process of assessing the frequency, duration, and extent of human exposure to air pollutants, focusing on hazardous air pollutants in this study. Exposure can be estimated using one of three approaches [58]: direct measurement, estimation, or exposure reconstruction. The proposed methodology utilizes the estimation approach.

In addition to the exposure scenarios and pathways, it is essential to consider the demographic and lifestyle factors that may influence exposure. These factors include age, gender, occupation, socioeconomic status, dietary habits, and cultural practices. Understanding these factors can help identify susceptible populations and inform targeted risk reduction strategies.

This section describes the equations used for estimating exposure, as well as exposure scenarios and locations and scenario-relevant exposure pathways. It is important to note that separate equations are used for inhalation and ingestion exposure to account for differences in exposure pathways, mechanisms, and potential health effects.

Exposure was estimated using Equations (2) and (3) [58]:

$$Exposure\ (inhalation) = \text{ADD (inhalation)} = \frac{C_{air} \times InhR \times ET \times EF \times ED}{\text{BW} \times AT} \tag{2}$$

where ADD represents the average daily dose (mg/kg-day), $C_{air}$ is the concentration of hazardous air pollutant(s) in the air (mg/m$^3$), $InhR$ is the inhalation rate (m$^3$/h), $ET$ is

the exposure time (h/day), *EF* is the exposure frequency (days/year), *ED* is the exposure duration (years), BW is the body weight (kg), and *AT* is the averaging time (days).

$$Exposure\ (ingestion) = \text{ADD (ingestion)} = \frac{C_{medium} \times IngR \times EF \times ED}{\text{BW} \times AT} \tag{3}$$

where ADD represents the average daily potential dose (mg/kg-day), $C_{medium}$ is the concentration of hazardous air pollutant(s) in the medium (e.g., mg/L, mg/g), *IngR* is the ingestion rate (e.g., L/day, g/day), *EF* is the exposure frequency (days/year), *ED* is the exposure duration (years), BW is the body weight (kg), and *AT* is the averaging time (days).

Exposure scenarios define the combination of exposure pathways and exposure parameters applied to complete the risk and hazard calculations described in the risk characterization section. For the purposes of this study, exposure scenarios include the resident adult and resident child scenarios [11], which differ in factors such as exposure duration, frequency, and body weight. The included exposure pathways are summarized in Table 5.

**Table 5.** Exposure Pathways Used in Multi-pathway Exposure Assessments [11].

| Exposure Pathways |
| :---: |
| Direct inhalation of vapors and particles |
| Incidental ingestion of soil |
| Ingestion of drinking water from treated surface water sources |
| Ingestion of homegrown produce [1,2] |

[1] Applies to farms in Wafra, Al-Abdali, and Al-Sulaibiya areas. [2] https://www.trade.gov/market-intelligence/kuwait-agriculture (accessed on 12 April 2023).

Exposure scenario locations refer to the physical places (i.e., receptors) within the study area chosen for evaluating residential adult and child exposure scenarios. To ensure thorough evaluation of all potential residential exposure locations throughout Kuwait, air dispersion modeling results were generated using two uniform Cartesian receptor grids and sensitive receptors. For more information on receptor grids and sensitive receptors, please refer to Section 2.2.4. This approach aligns with established best practices in the field [11]. Figure 2 illustrates the relationship between human exposure to HAPs through multiple pathways. HAPs exist in various environmental media, such as air, water, and soil, and can enter the human body via different exposure routes, including ingestion and inhalation. HAPs are deposited into environmental media, such as water and soil, through atmospheric deposition.

2.3.3. Quantitative Estimation of Cancer Risk and Non-cancer Hazard (Step 6 in the Proposed Methodology)

Risk characterization is a crucial component in human health risk assessments. It synthesizes information from preceding assessment steps, such as transport-and-fate modeling (Section 2.3.1) and exposure quantification (Section 2.3.2), to determine the presence or absence of risk and quantify expected outcomes. Furthermore, risk characterization identifies key factors contributing to risk, enabling regulators, engineers, analysts, and stakeholders to make informed decisions about appropriate actions for risk reduction. It is essential to recognize the uncertainties and limitations inherent in the risk characterization process, which may stem from factors like incomplete emissions data, variability in exposed populations, and assumptions made during the assessment to account for temporal variability in emissions. Acknowledging these uncertainties provides context for the results and informs future research or decision making.

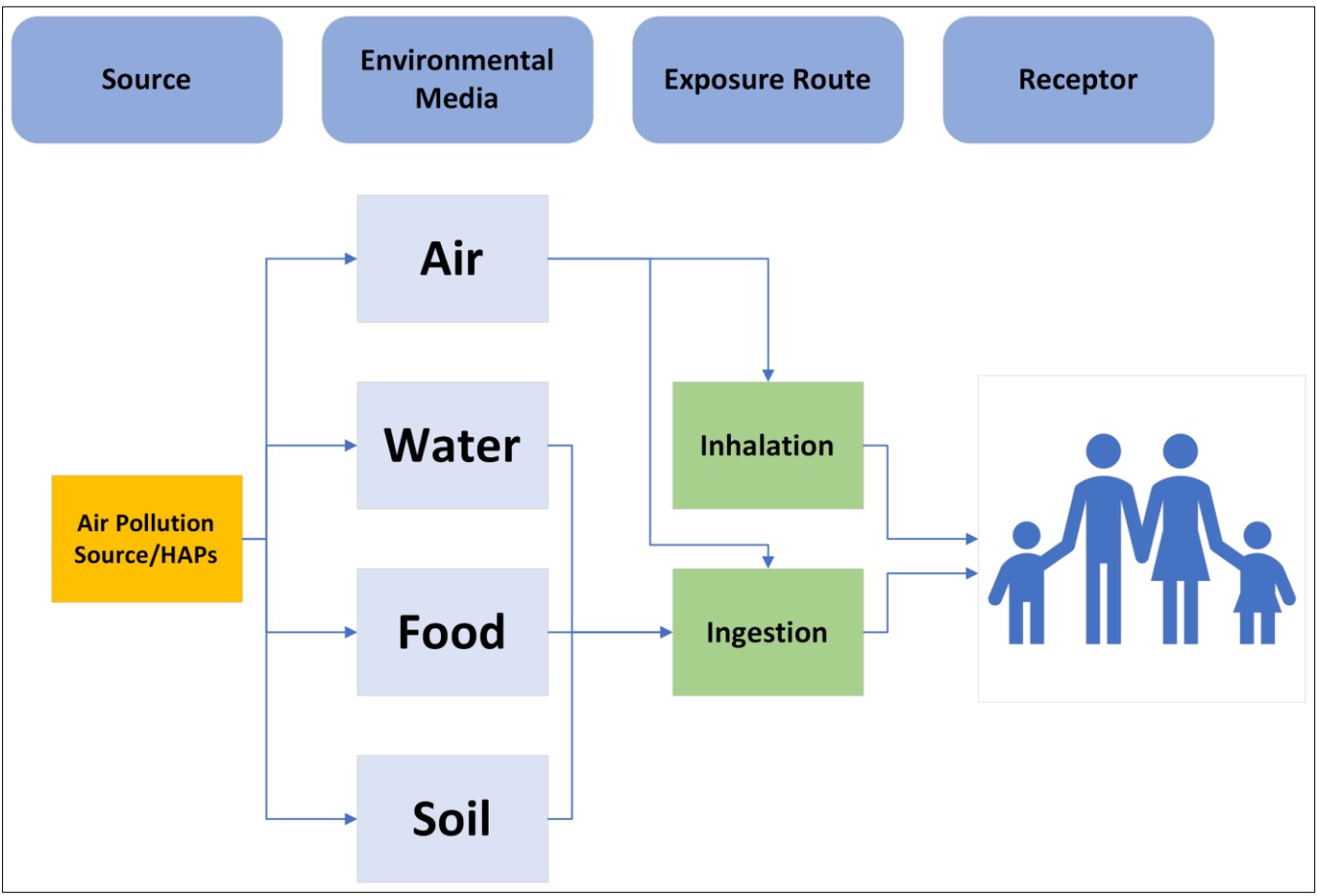

**Figure 2.** A visual representation of the Multi-pathway Human Exposure to Hazardous Air Pollutants.

Cancer risk is the incremental probability of an individual developing cancer over a lifetime due to exposure to a specific carcinogenic hazardous air pollutant or a mixture of carcinogenic hazardous air pollutants [11]. The predicted risk represents the additional risk of cancer from the exposure being analyzed, which is supplementary to other risks resulting from numerous factors (e.g., smoking). This risk is expressed as a probability ranging from zero (no risk) to one (certainty of developing cancer), without any units or dimensions. Cancer risk is represented as a numerical value, such as "1 in 100,000" or "$1 \times 10^{-5}$", indicating that one individual in a population of 100,000 is expected to develop cancer due to exposure to the carcinogenic hazardous air pollutant over a lifetime. The calculation of cancer risk considers several factors, such as the concentration of the carcinogenic hazardous air pollutant in the environment, exposure duration, frequency, and exposure route (e.g., ingestion or inhalation), as well as the hazardous air pollutant's cancer potency factor or slope factor.

For inhalation exposures, chronic cancer risks for individual pollutants are estimated by multiplying the long-term exposure concentration estimate by the corresponding pollutant-specific inhalation unit risk. This calculation determines the potential incremental cancer risk, as illustrated in Equation (4):

$$Cancer\ Risk\ (inhalation) = Inhalation\ Exposure\ Concentration \times Inhalation\ Unit\ Risk \quad (4)$$

For each ingestion exposure pathway being evaluated, chronic cancer risks for individual pollutants are estimated by multiplying the lifetime average daily dose estimate

for each ingestion exposure pathway by the pollutant-specific cancer slope factor. This calculation estimates potential incremental cancer risk, as illustrated in Equation (5):

$$Cancer\ Risk\ (ingestion) = Lifetime\ Average\ Daily\ Dose \times Cancer\ Slope\ Factor \quad (5)$$

Non-cancer hazard refers to potential adverse non-cancer health effects, such as respiratory issues, neurological complications, and neurotoxicity, resulting from exposure to a hazardous air pollutant or a mixture of hazardous air pollutants. Exposure duration plays a significant role in hazard classification, with two primary categories: acute hazards associated with short-term exposure and chronic hazards linked to long-term exposure. Evaluating non-cancer hazards involves comparing the estimated exposure level of a substance to its reference dose or reference concentration, which represent levels deemed safe without appreciable risk of adverse health effects throughout a lifetime of exposure [59]. The non-cancer hazard is calculated using Equation (6):

$$Hazard\ Quotient = \frac{Estimated\ Exposure\ Level}{Reference\ Concentration\ (or\ Reference\ Dose)} \quad (6)$$

The hazard index, a conservative measure, is used in human health risk assessments involving multiple hazardous air pollutants or exposure pathways. This index represents the sum of the individual hazard quotients for each hazardous air pollutant or pathway.

In this research, we define cumulative risk (cancer or non-cancer) as the risk resulting from simultaneous exposure to multiple hazardous air pollutants from multiple sources through one or more pathways or exposure routes, such as inhalation or ingestion. The total cumulative cancer risk was computed using Equation (7):

$$R_{total} = \sum (R_{i,j}) + \sum (R_{i,inhalation}) \quad (7)$$

where $R_{total}$ represents the cumulative cancer risk estimate, considering both ingestion and inhalation pathways, $R_{i,j}$ represents the individual risk estimate for hazardous air pollutant i and ingestion pathway j, $R_{i,inhalation}$ represents the individual risk estimate for hazardous air pollutant i through the inhalation pathway, i denotes the different hazardous air pollutants (e.g., hazardous air pollutant 1, hazardous air pollutant 2, hazardous air pollutant 3, etc.), and j denotes the different ingestion pathways (e.g., vegetable ingestion, fish ingestion, egg ingestion, beef ingestion, etc.). On the other hand, the total cumulative non-cancer risk (hazard index) was estimated using Equation (8):

$$Hazard\ Index_{total} = \sum (HQ_{i,j}) + \sum (HQ_{i,inhalation}) \quad (8)$$

where $Hazard\ Index_{total}$ represents the total hazard index, considering both ingestion and inhalation pathways, $HQ_{i,j}$ represents the hazard quotient for hazardous air pollutant i and ingestion pathway j, $HQ_{i,inhalation}$ represents the hazard quotient for hazardous air pollutant i through the inhalation pathway, i denotes the different hazardous air pollutants (e.g., hazardous air pollutant 1, hazardous air pollutant 2, hazardous air pollutant 3, etc.), and j denotes the different ingestion pathways (e.g., vegetable ingestion, fish ingestion, egg ingestion, beef ingestion, etc.).

Lastly, the acute inhalation hazard quotient was calculated using Equation (9):

$$Acute\ Inhalation\ Hazard\ Quotient = \frac{acute\ air\ concentration}{acute\ inhalation\ exposure\ criteria} \quad (9)$$

Regulatory agencies establish target risk levels to assess potential health risks from hazardous air pollutant exposure and ensure public safety. Carcinogenic risk levels typically range from $1 \times 10^{-4}$ to $1 \times 10^{-6}$, indicating a negligible increase in cancer risk due to exposure. For non-cancer health effects, a hazard quotient or hazard index of 1 or lower indicates no expected adverse effects, while values above 1 suggest potential risks.

In this study, based on the U.S. EPA Region 6 HHRAP Addendum [60], we adopted narrower target risk levels for hazardous air pollutant exposure to account for conservatism and background concentration exposure within the study area. Consequently, cancer risk levels should not exceed $1 \times 10^{-5}$, and the hazard quotient/index should not exceed 0.25.

It is essential to note that target risk levels are not definitive indicators of adverse effects, as risk estimations inherently have uncertainties. These uncertainties can arise from exposure variability, limited toxicological data, temporal emissions' variability, and assessment assumptions. If calculated risks are within target values, no further investigation is needed, and the evaluated conditions are not considered to present unacceptable risks. However, when target values are exceeded, the methodology entails further review by risk managers, risk analysts, or engineers. This review process involves examining the scientific basis and uncertainties associated with the calculation, including toxicity factors and uncertainty factors addressing study limitations and data quality issues, as well as investigating the contributing factors and underlying drivers of unacceptable risks.

2.3.4. Risk Driver Analysis: Forensic Analysis of Human Health Risks (Step 7 in the Proposed Methodology)

Risk driver analysis is a systematic, project-specific process that addresses the fundamental question, "Where does the risk come from?". It is important to identify and understand the primary factors contributing to unacceptable cancer and non-cancer risks associated with hazardous air pollutant exposure in a specific project setting. This stage involves an examination of various exposure elements, including scenarios, such as farmers, residents, local fish consumers, construction workers, and outdoor workers, as well as pathways like air, water, soil, plants, and animals.

The analysis assesses the significance of specific exposure pathways, such as inhalation, ingestion, and dermal contact, to pinpoint the most critical factors leading to elevated risk levels. A vital aspect of this process is tracing back to the sources or chemicals contributing to the increased risks. Additionally, the analysis investigates the potential for bioaccumulation of hazardous air pollutants in tissues or agricultural produce to enhance understanding of the risks associated with human health in the project area. For instance, methylmercury is a lipophilic substance that tends to bioaccumulate in the food chain [61,62].

The project-specific risk assessment is re-evaluated upon completing the analysis to ensure accuracy and guide development of targeted risk reduction strategies. These strategies may include emission controls, exposure reduction measures, or regulatory actions aimed at addressing the identified risk drivers and protecting public health within the project's scope. Figure 3 illustrates a simplified example of a risk driver analysis. As shown in the example, the risk driver analysis employs a systematic and methodological approach to investigate the various components of human health risk assessment, identifying the elements contributing to elevated risk and tracing them back to the emission sources or hazardous air pollutants driving the risk.

By implementing this methodology, we effectively evaluated human health risks associated with exposure to hazardous air pollutants in Kuwait. The methodology encompasses steps such as developing emissions inventories for each air quality zone, conducting air dispersion modeling using AERMOD, estimating pollutant concentrations at the point of contact, identifying exposure scenarios, and assessing levels and frequencies of contact between humans and pollutants.

Through the process of calculating multi-pathway and cumulative risks, our assessment considered both direct and indirect exposure pathways. This allowed us to examine the contributing factors and underlying drivers of unacceptable risks in each air quality zone, ultimately providing a consistent and robust risk characterization upon which risk mitigation strategies can be developed to protect human health in Kuwait.

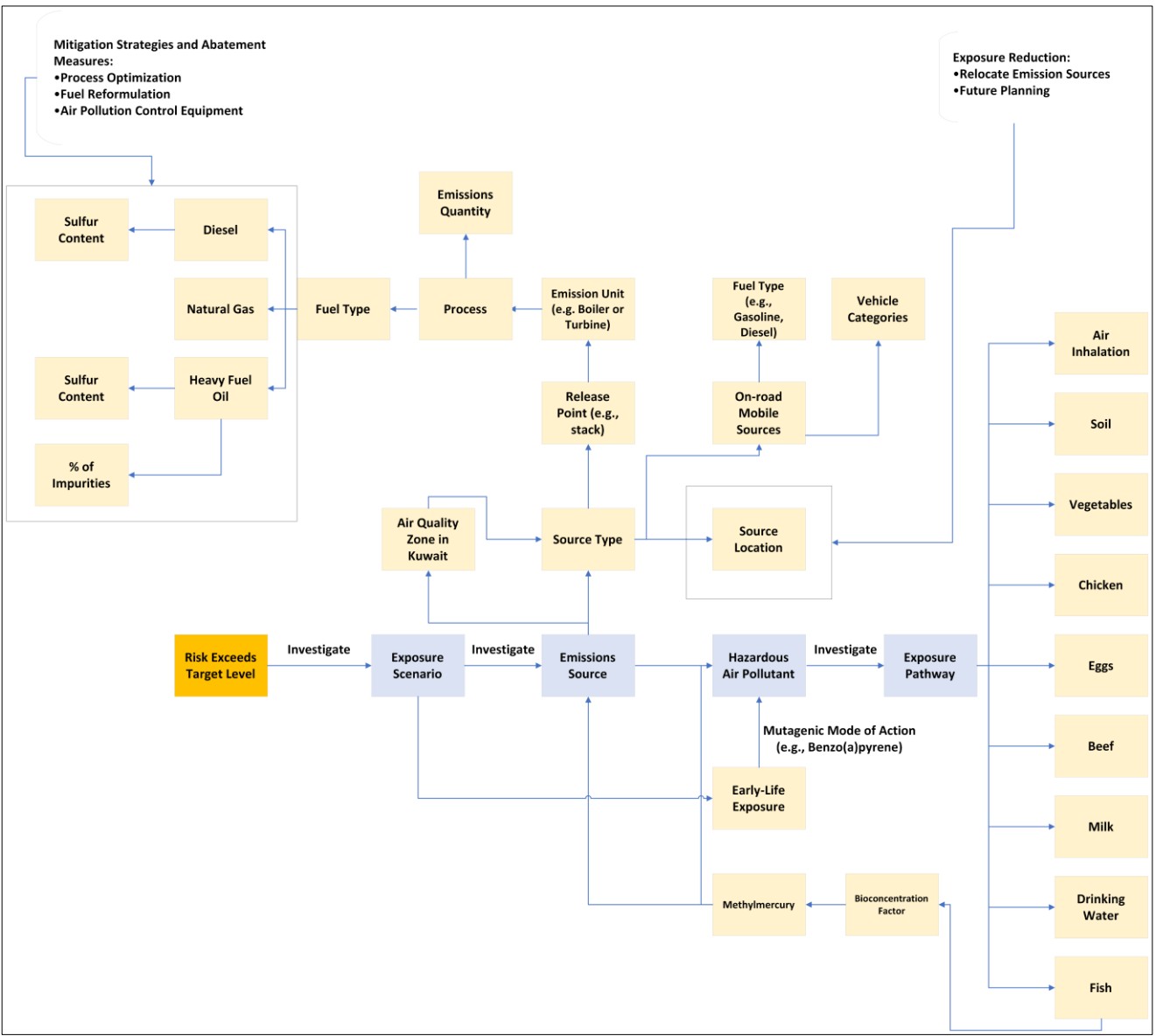

**Figure 3.** Simplified Example of a Risk Driver Analysis.

### 3. Results

*3.1. Case Study*

This section presents the application of the human health risk assessment methodology outlined above to a focused case study in Kuwait, using data from the year 2017. The analysis concentrates on hazardous air pollutants, emphasizing the impact of direct emissions from industrial sources across Kuwait's three air quality zones. Although mobile sources and fugitive emissions are significant sources of localized hazardous air pollutants [12,55,63], they were intentionally excluded from this case study. This approach enables us to concentrate on the effects of industrial emissions, providing a clear baseline for the analysis. In addition, industrial sources are regulated which provides a legal framework upon which emissions reduction actions can be developed to address specific issues identified through the health risk analysis process.

Emission sources, representative of Kuwait's distinctive pollution landscape, were selected to ensure an accurate depiction. Our selection employed a multi-criteria approach, factoring in not only the type of industry, geographical location within air quality zones, and the intensity of emissions but also their unique contribution to Kuwait's pollution landscape.

The unique contribution refers to how certain industries in Kuwait are characterized by specialized processes and materials, resulting in distinct emissions profiles and pollution effects, which differ from other regions. In addition, we further enriched our criteria by considering the proximity to populated areas, meteorological conditions, and land-use characteristics (inland versus shoreline), acknowledging their potential impact on the dispersion and intensity of pollutants. This multi-faceted selection process ensures that our study is comprehensive, representative, and aligned with the actual pollution landscape in Kuwait.

The results of our assessments for each air quality zone, to be presented later in this paper, provide an understanding of the prevailing air quality conditions and associated health risks in Kuwait. This case study effectively underscores the practical applicability and potential impact of our proposed methodology in human health risk assessment. A map of the study region is provided in Figure 4 for a clearer geographical context.

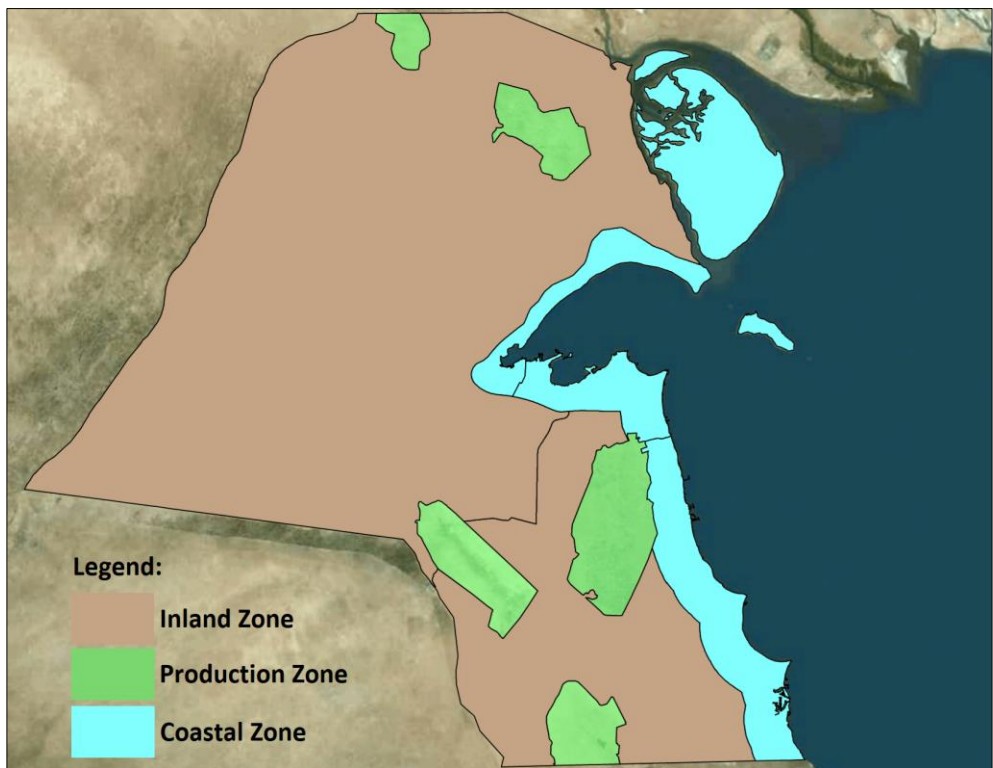

**Figure 4.** Kuwait Air Quality Zones. Map Data: Lakes Satellite.

In our efforts to accurately represent Kuwait's unique pollution landscape, we selected several facilities of interest across the three air quality zones, each characterized by different types of emission sources. Table 6 presents a summary of the number of modeled facilities in each air quality zone, along with the number of their corresponding emission sources. Additionally, Table 7 lists the hazardous air pollutants that were considered in our study.

**Table 6.** Summary of the Number of Modeled Facilities and Corresponding Number of Emission Sources in the Air Quality Zones of Kuwait.

| Air Quality Zone | Number of Modeled Facilities | Number of Emission Sources within Modeled Facilities |
|---|---|---|
| Inland | 4 | 21 |
| Production | 7 | 21 |
| Coastal | 10 | 47 |

**Table 7.** Hazardous Air Pollutants Considered in the Study.

| Hazardous Air Pollutants |
| :---: |
| Benzene |
| Formaldehyde |
| Toluene |
| Benzo(a)pyrene |

For this study, we selected a small, representative subset of hazardous air pollutants known for their adverse impacts on human health and their common presence in industrial emissions. The selected pollutants, drawn from a larger emissions inventory, have been chosen due to their potency, their prevalence in industrial emissions, and the extensive availability of both cancer and non-cancer assessments in the literature. Furthermore, a subset of pollutants is all that is required in order to demonstrate that by using a normalized pollutant that the human health impacts from any number of hazardous air pollutants can be easily found, meaning only one calculation of the time-consuming air dispersion modeling is required. By limiting the pollutants analyzed, this paper does not provide a comprehensive picture of air pollutant health risks in Kuwait; such a comprehensive impact assessment is outside the focus and scope of this paper. Table 1 (refer to Section 1.1) details the adverse health effects associated with these selected pollutants.

*3.2. Cumulative Risk Results*

This subsection presents the cumulative health risks from all pathways evaluated within our human health risk assessment study. This includes the analysis of cancer, non-cancer (hazard index), and acute non-cancer health effects within Kuwait's three distinctive air quality zones: inland, production, and coastal. For a detailed definition of cancer risk and non-cancer risk, please refer to Section 2.3.3 of this paper. In the tables presented below, we provide numerical data to illustrate the calculated health risks for each receptor, under specific exposure scenarios, in the inland, production, and coastal air quality zones. Each value represents a statistical estimate of the risk posed to a "receptor", which refers to an individual, either an adult or a child, residing in the area. These risk estimates are calculated based on potential exposure to various air pollutants, considering factors such as pollutant concentrations, exposure duration, frequency, and the inherent toxicity of the pollutants.

3.2.1. Inland

Tables 8–10 provide a summary of the cumulative cancer risk, cumulative non-cancer risk, and cumulative acute non-cancer risk for the inland air quality zone. The sensitive receptors for this zone were identified based on the grid nodes, which exhibited the maximum air concentration values. This approach ensures the analysis focuses on the areas with the highest potential exposure.

**Table 8.** Cumulative Cancer Risk.

| Sensitive Receptor/Exposure Scenarios | Resident Adult | Resident Child |
| :---: | :---: | :---: |
| Receptor 1 | $4.79 \times 10^{-11}$ | $9.63 \times 10^{-12}$ |
| Receptor 2 | $4.74 \times 10^{-9}$ | $9.55 \times 10^{-10}$ |
| Receptor 3 | $1.02 \times 10^{-8}$ | $2.05 \times 10^{-9}$ |
| Receptor 4 | $3.77 \times 10^{-11}$ | $1.76 \times 10^{-11}$ |

**Table 9.** Cumulative Non-cancer Risk.

| Sensitive Receptor/Exposure Scenarios | Resident Adult | Resident Child |
|---|---|---|
| Receptor 1 | $8.83 \times 10^{-7}$ | $8.94 \times 10^{-7}$ |
| Receptor 2 | $9.30 \times 10^{-5}$ | $9.86 \times 10^{-5}$ |
| Receptor 3 | $2.02 \times 10^{-4}$ | $2.16 \times 10^{-4}$ |
| Receptor 4 | $3.45 \times 10^{-8}$ | $8.93 \times 10^{-8}$ |

**Table 10.** Cumulative Acute Non-cancer Risk.

| Sensitive Receptor/Exposure Scenarios | Resident Adult/Child |
|---|---|
| Receptor 1 | $4.47 \times 10^{-5}$ |
| Receptor 2 | $2.21 \times 10^{-4}$ |
| Receptor 3 | $3.00 \times 10^{-4}$ |
| Receptor 4 | $8.61 \times 10^{-10}$ |

The inland zone demonstrates a low health risk profile, with all sensitive receptors displaying risks below the established thresholds, as outlined in Section 2.3.3, for cancer, non-cancer, and acute non-cancer effects. The identified pollutants do not appear to pose significant health hazards for either adults or children within this air quality zone, based on our receptor selection.

### 3.2.2. Production

Tables 11–13 provide a summary of the cumulative cancer risk, cumulative non-cancer risk, and cumulative acute non-cancer risk for the production air quality zone. Consistent with the inland air quality zone, the sensitive receptors for this zone were determined based on the grid nodes with the calculated maximum air concentration values. Though these receptors share the same designations as those in the inland zone, they represent distinct locations within the production air quality zone. This method of selection maintains the focus of the analysis on areas with the highest potential exposure.

**Table 11.** Cumulative Cancer Risk.

| Sensitive Receptor/Exposure Scenarios | Resident Adult | Resident Child |
|---|---|---|
| Receptor 1 | $3.11 \times 10^{-11}$ | $6.26 \times 10^{-12}$ |
| Receptor 2 | $2.34 \times 10^{-9}$ | $4.73 \times 10^{-10}$ |
| Receptor 3 | $1.20 \times 10^{-8}$ | $2.41 \times 10^{-9}$ |
| Receptor 4 | $3.83 \times 10^{-11}$ | $1.76 \times 10^{-11}$ |

**Table 12.** Cumulative Non-cancer Risk.

| Sensitive Receptor/Exposure Scenarios | Resident Adult | Resident Child |
|---|---|---|
| Receptor 1 | $5.76 \times 10^{-7}$ | $5.86 \times 10^{-7}$ |
| Receptor 2 | $4.45 \times 10^{-5}$ | $4.64 \times 10^{-5}$ |
| Receptor 3 | $2.38 \times 10^{-4}$ | $2.55 \times 10^{-4}$ |
| Receptor 4 | $3.39 \times 10^{-8}$ | $8.85 \times 10^{-8}$ |

**Table 13.** Cumulative Acute Non-cancer Risk.

| Sensitive Receptor/Exposure Scenarios | Resident Adult/Child |
|---|---|
| Receptor 1 | $2.49 \times 10^{-5}$ |
| Receptor 2 | $2.71 \times 10^{-4}$ |
| Receptor 3 | $2.71 \times 10^{-4}$ |
| Receptor 4 | $8.78 \times 10^{-10}$ |

Our study findings for the production zone align closely with those from the inland zone. No significant health risks were identified, as all cumulative risks across different population segments remained below the specified thresholds. This finding indicates that the current levels of selected pollutants may not contribute significantly to health impacts within the production zone.

3.2.3. Coastal

Tables 14–16 present a summary of the cumulative cancer risk, cumulative non-cancer risk, and cumulative acute non-cancer risk within the coastal air quality zone. The sensitive receptors identified in this zone have been selected due to their strategic placement within densely populated areas and locations where populations are likely to be more susceptible to adverse effects from exposure to hazardous air pollutants. Such locations include critical community hubs such as hospitals, schools, resorts, and places of worship. The chosen receptors are of particular interest due to the intersecting factors of high potential pollutant concentrations and significant population presence, which jointly contribute to an increased potential for exposure.

**Table 14.** Cumulative Cancer Risk.

| Sensitive Receptor/Exposure Scenarios | Resident Adult | Resident Child |
|---|---|---|
| General Ahmadi Hospital | $6.12 \times 10^{-6}$ | $2.31 \times 10^{-6}$ |
| Fatima Bint Asad High School for Girls | $7.56 \times 10^{-6}$ | $2.87 \times 10^{-6}$ |
| **Ahmadi Hospital** | $\mathbf{1.09 \times 10^{-5}}$ | $4.33 \times 10^{-6}$ |
| Ahmadi Zoo | $7.92 \times 10^{-6}$ | $3.04 \times 10^{-6}$ |
| Adan Hospital | $4.05 \times 10^{-6}$ | $1.52 \times 10^{-6}$ |
| Hilton Kuwait Resort | $3.63 \times 10^{-6}$ | $1.31 \times 10^{-6}$ |
| Mosque (North Ahmadi) | $5.79 \times 10^{-6}$ | $2.23 \times 10^{-6}$ |

**Table 15.** Cumulative Non-cancer Risk.

| Sensitive Receptor/Exposure Scenarios | Resident Adult | Resident Child |
|---|---|---|
| General Ahmadi Hospital | $9.67 \times 10^{-3}$ | $1.56 \times 10^{-2}$ |
| Fatima Bint Asad High School for Girls | $1.19 \times 10^{-2}$ | $1.93 \times 10^{-2}$ |
| Ahmadi Hospital | $1.26 \times 10^{-2}$ | $2.43 \times 10^{-2}$ |
| Ahmadi Zoo | $1.13 \times 10^{-2}$ | $1.92 \times 10^{-2}$ |
| Adan Hospital | $6.02 \times 10^{-3}$ | $9.92 \times 10^{-3}$ |
| Hilton Kuwait Resort | $6.84 \times 10^{-3}$ | $1.00 \times 10^{-2}$ |
| Mosque (North Ahmadi) | $7.76 \times 10^{-3}$ | $1.36 \times 10^{-2}$ |

**Table 16.** Cumulative Acute Non-cancer Risk.

| Sensitive Receptor/Exposure Scenarios | Resident Adult/Child |
|---|---|
| General Ahmadi Hospital | $1.19 \times 10^{-1}$ |
| Fatima Bint Asad High School for Girls | $1.16 \times 10^{-1}$ |
| Ahmadi Hospital | $9.69 \times 10^{-2}$ |
| Ahmadi Zoo | $1.05 \times 10^{-1}$ |
| Adan Hospital | $6.81 \times 10^{-2}$ |
| Hilton Kuwait Resort | $9.65 \times 10^{-2}$ |
| Mosque (North Ahmadi) | $8.29 \times 10^{-2}$ |

The cumulative health risks in the coastal air quality zone are within acceptable limits, however, they are notably higher when compared to our findings for the inland and production zones. For cancer risks, only the Ahmadi Hospital, highlighted in boldface in Table 14, with a cancer risk of $1.09 \times 10^{-5}$ for the resident adult exposure scenario, slightly exceeds the specified threshold of $1 \times 10^{-5}$.

To provide a practical understanding of the cancer risk value above, consider it in terms of population-scale risk. If 100,000 adults were exposed to the same level of carcinogenic hazardous air pollutants—such as benzene, formaldehyde, toluene, and benzo(a)pyrene—from multiple emission sources over a lifetime, approximately 1.09 individuals in this group could be expected to develop cancer due to this exposure. However, this is a statistical estimate and does not guarantee that any particular individual in this group will indeed develop cancer from this exposure.

To further contextualize, the cancer risk at the Ahmadi Hospital sensitive receptor can be roughly compared to the risk of dying from an encounter with a hornet, wasp, or bee sting, which is 1 in 54,516 [64]. This comparison serves to convey that, while non-zero, the additional risk posed by exposure to these pollutants is relatively manageable when viewed alongside everyday risks. However, it should be remembered that these risk assessments rely on models and certain assumptions, including constant, long-term exposure. Actual risk may vary depending on factors such as genetic predispositions, lifestyle choices, and actual exposure patterns.

For the non-cancer and acute non-cancer risks, all sensitive receptors displayed values below the defined thresholds of 0.25. This result suggests that, while the level of pollutants in the coastal air quality zone is higher, it does not exceed the threshold that would pose significant health risks to the population. However, the risk is more pronounced in this zone compared to the inland and production zones.

## 4. Discussion

In this section, we analyze the key risk drivers, including specific pollutants, their sources, and the main exposure pathways. Subsequently, we consider the strengths and limitations of our methodology. Finally, we conclude by examining the variability and uncertainty associated with our analysis.

### 4.1. Risk Driver Analysis

The risk driver analysis from our assessment identified a singular case of cumulative cancer risk exceeding the target risk levels in the coastal air quality zone. This elevated risk was specifically noted at Ahmadi Hospital for the resident adult exposure scenario. Our systematic examination of exposure scenarios, emission sources, and exposure pathways indicated that the primary risk driver was benzo(a)pyrene. Moreover, our analysis suggested that the risk is primarily associated with indirect exposure pathways.

This finding implies that the origin of this pollutant may be localized emissions within the coastal zone, potentially originating from industrial activities or combustion processes. Due to confidentiality restrictions, specific information identifying the sources cannot be disclosed. However, it is worth noting that the likely culprits include industrial flares and other combustion sources. Consequently, the insights provided by our method serve as a key tool for policymakers and regulators. Our findings can guide the consideration and implementation of targeted risk reduction strategies, such as emission control measures at identified sources or various mitigation strategies. The ultimate goal of these strategies, informed by our research, would be to uphold public health standards within the project's scope while reducing the cancer risk to acceptable levels.

### 4.2. Strengths and Limitations

In this section, we explore the strengths and limitations of the research methodology employed in this case study. A key strength is the integration of previously validated methods to quantify the country-wide cumulative multi-pathway human health impacts associated with exposure to hazardous air pollutants in Kuwait. While these methods have been previously validated, our research combines them in a unique manner to address the specific challenges of assessing cumulative, multi-pathway risks in a region with complex industrial activities. Additionally, the proposed methodology is flexible in incorporating more sophisticated multi-layer, multi-species air dispersion models, such as non-steady-

state puff models like CALPUFF or SCIPUFF. The advantage of non-steady-state puff models over steady-state plume models lies in their capacity to capture the temporal and spatial variability in meteorological conditions, leading to a more accurate representation of pollutant dispersion patterns, thereby improving the precision and reliability of the exposure and risk estimates.

However, we must also acknowledge certain limitations of our approach. First, building downwash effects were not evaluated due to lack of available parameterization. Second, our study did not account for long-range transport of pollutants from areas outside of Kuwait.

### 4.3. Variability and Uncertainty

Variability and uncertainty in human health risk assessments originate from biological differences, external environmental factors, and the presence of gaps in both data availability and scientific knowledge. Variability refers to the differences or range of responses to environmental exposure among individuals or populations and can be attributed to factors such as genetics, age, lifestyle, and pre-existing health conditions. Environmental factors, such as the duration and route of exposure, as well as spatial–temporal variations in hazardous air pollutant concentrations, further contribute to this variability.

On the other hand, uncertainty refers to the lack or incompleteness of data regarding the input parameters for the methods employed to evaluate human health impacts. This form of uncertainty is inherent at every stage of the assessment process, from limitations in data availability to the extrapolation of results from animal studies to humans. Enhancing the reliability of risk assessments requires effectively addressing both variability and uncertainty. This can be accomplished by employing tools such as stochastic modeling and sensitivity analysis and by emphasizing the necessity for continuous research, monitoring, and data collection to reduce uncertainty over time.

### 5. Conclusions

This paper proposes an innovative approach to quantify country-wide cumulative risk (cancer, non-cancer, and acute) resulting from simultaneous exposure to multiple hazardous air pollutants from multiple sources through one or more pathways or exposure routes, such as direct inhalation or indirect ingestion. The authors also introduce a new method for identifying chemical and pathway risk drivers which provides a deeper understanding of potential health risks and provides a starting point for targeted risk reduction. The conclusions drawn from this work are as follows:

1.  The overall health risk profile across the inland, production, and coastal air quality zones of Kuwait is low to moderate, as most risk values lie beneath the established risk thresholds. This suggests that the current levels of pollutants quantified in this case study do not likely pose significant health threats to the adult and child residential population.
2.  The coastal air quality zone has a higher risk profile compared to the inland and production zones, particularly for cancer risks. However, these values are mostly within acceptable limits. An exception is the Ahmadi Hospital for the resident adult exposure scenario, where the cancer risk slightly exceeds the target level.
3.  The risk driver analysis identified benzo(a)pyrene as the primary risk driver contributing to the elevated cancer risk at the Ahmadi Hospital in the coastal zone, calculated to be $1.09 \times 10^{-5}$, with the likely sources being local industrial emissions or combustion processes. However, it is important to note that our analysis assumes a resident adult exposure scenario that may not accurately represent the real-world exposure at the hospital. Specifically, it is unlikely that an individual would be exposed 24 h a day, 7 days a week over a lifetime in this location, suggesting that our estimates might overstate the actual risk.

4.  The inherent variability and uncertainty in the risk estimates are recognized, emphasizing the need for careful interpretation and further research, such as the employment of stochastic modeling.

This study improves our understanding of potential health impacts from exposure to hazardous air pollutants in this region, providing valuable insights for future environmental management strategies and policy decisions. Moreover, the benefits of this research, such as forecasting health impacts, could be utilized to optimize regulatory-based permitting, public education, emergency room care, and availability of response specialists, with the end goal of improving overall healthcare management that directly considers the health impacts of air quality on the exposed population of Kuwait.

**6. Future Work**

The proposed methodology, as presented, sets the stage for future work. In this phase, a stochastic model will be developed, incorporating probabilistic distributions of variables such as body weight, inhalation rate, metabolic rates, and ingestion rates. This approach will address variability and uncertainty in estimating risks and will be made possible through the development and implementation of the proposed methodology. Additionally, Step 3 of the proposed methodology, which involves estimating the concentrations of hazardous air pollutants at the point-of-contact for receptor populations by conducting an environmental transport-and-fate analysis, will be used in part to estimate the ecological screening quotient for country-wide ecological risk assessments. This assessment will also consider the potential pathways through which ecological changes might affect human health, such as impacts on food chains, water quality, and biodiversity. By integrating these ecological aspects with the direct human health risk factors, the methodology aims to provide a more holistic approach to both human health and ecological integrity.

**Author Contributions:** Conceptualization, M.M., J.V.G.T., and R.F.; Data curation, M.M., J.V.G.T., and A.R.; Formal analysis, M.M.; Funding acquisition, J.V.G.T.; Investigation, M.M.; Methodology, M.M. and A.R.; Project administration, J.V.G.T. and R.F.; Resources, J.V.G.T. and R.F.; Software, M.M. and J.V.G.T.; Supervision, J.V.G.T. and R.F.; Validation, M.M.; Visualization, M.M.; Writing—original draft, M.M.; Writing—review and editing, M.M., J.V.G.T., R.F., and B.M. All authors have read and agreed to the published version of the manuscript.

**Funding:** This research and the APC were funded by Lakes Environmental Software.

**Informed Consent Statement:** Not applicable.

**Data Availability Statement:** A subset of the data is available upon request to the corresponding author, due to confidentiality constraints on certain portions of the dataset.

**Acknowledgments:** The authors would like to thank Jeff Stoakes from the Air Toxics Risk Assessments department at the Office of Air Quality, Indiana Department of Environmental Management, for his helpful comments and review of the manuscript.

**Conflicts of Interest:** The authors declare no conflict of interest.

**Abbreviations and Nomenclature**

| Abbreviations/Nomenclature | Meaning |
| --- | --- |
| $\mu g/m^3$ | micrograms per cubic meter |
| ADD | average daily dose (inhalation, ingestion) |
| API | American Petroleum Institute |
| AQZ | air quality zone |
| AT | averaging time |
| BTEX | benzene, toluene, ethylbenzene, and xylene |
| BW | body weight |

| $C_{air}$ | concentration of pollutant(s) in the air |
| $C_{medium}$ | concentration of pollutant(s) in the medium |
| ED | exposure duration |
| EF | exposure frequency |
| ET | exposure time |
| g | gram |
| HAPs | hazardous air pollutants |
| HHRAP | Human Health Risk Assessment Protocol |
| *IngR* | ingestion rate |
| *InhR* | inhalation rate |
| ISCST3 | Industrial Source Complex Short Term 3 |
| L | liter |
| mg | milligram |
| NASA | National Aeronautics and Space Administration |
| PAHs | polycyclic aromatic hydrocarbons |
| SRTM | Shuttle Radar Topography Mission |
| TFER | transport, fate, exposure, and risk characterization |
| U.S. EPA | U.S. Environmental Protection Agency |
| VOCs | volatile organic compounds |
| WRF | Weather Research and Forecasting |

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
