# Peer review of "Extending Multi-Pathway Human Health Risk Assessment from Regional to Country-Wide—A Case Study on Kuwait"

_atmosphere, doi:10.3390/atmos14081247_

Round 1
Reviewer 1 Report
This study conducts a human health risk assessment in Kuwait using methods developed by the US EPA. Although the paper is written in an easy-to-understand manner, and the fact that it targets a region where few human health risk assessments have been conducted is commendable, I regret that it seems difficult to publish this as an academic paper for the following reasons.
(1) The method developed by the US EPA is used and is not novel.
(2) If a large number of substances (e.g., hundreds to thousands), which is the advantage of this method, were actually evaluated, the paper would still be worthy of publication, but only four substances were actually covered, which is clearly insufficient to grasp the full picture of Kuwait's health risks.
Reviewer 2 Report
The manuscript is well prepared and the scientific approach is of high interest as well as complete and well done.
The proposed methodology can be a point of reference to be followed in other locations.
There are no suggestions for changes in the manuscript.
As a suggestion, the attached document shows the analysis for coincidences with the published works. It is advised to put attention to not making alike texts in some paragraphs and change those with long similar sentences.

The writing is adequate and there are not significant changes to be done.
Author Response
Thank you.
Reviewer 3 Report
The manuscript propose a comprehensive method to assess the human health risk based on a number of factors including pollutants, transport/dispersion, and meteorological conditions. However, there are issues that need to be addressed before further consideration.
1. This manuscript is a research article, not a review; thus, it should be shortened to focus on the most significant points.
2. The proposed method includes many results obtained by modelling/simulation. How can the authors evaluate these results?
3. A similar problem is also present in the results of risk assessment. How can the authors prove that these results are reliable?
Round 2
Reviewer 1 Report
My biggest concern about this paper was originality, but the authors have made significant revisions to the manuscript to show the additional scientific value of this paper. The authors have also made appropriate supplementary descriptions of the risk assessment results to avoid misunderstandings. Based on the above, I believe that my concerns have been addressed and the paper is acceptable for publication.
Reviewer 3 Report
The authors have revised the manuscript appropriately, thus, my suggestion is that it can be accepted with minor revision for proofreading. It is a bit in a muddle with track-change version. For example, page 4 of 28.